

# The impact of the military conflict in Sudan on maternal health: a mixed qualitative and quantitative study

Elhadi Miskeen

Department of Obstetrics and Gynecology, College of Medicine, University of Bisha, Bisha, Saudi Arabia

## ABSTRACT

**Objectives:** Ongoing military conflict in Sudan has had significant repercussions on the health and well-being of the population, particularly among women of reproductive age. This study aimed to investigate the impact of conflict on maternal health by employing a mixed qualitative and quantitative research approach.

**Methods:** Through in-depth interviews and survey questionnaires (388 women), this study examined the experiences and challenges faced by pregnant women and new mothers and the availability and accessibility of maternal healthcare services in conflict-affected areas. Using a qualitative approach, in-depth interviews were conducted with 35 women who had recently given birth or were pregnant in regions affected by the Khartoum State–Sudan conflict. Thematic analysis was used to analyze the data collected from the interviews.

**Result:** Most women did not have access to healthcare services (86.6%), and out of the total sample, 93 (24%) experienced adverse outcomes. The factors associated with adverse effects were parity (OR 1.78, CI [1.15–2.75], $p$-value 0.010), gestational age (OR 2.10, CI [1.36–3.25], $p$-value 0.002), access to healthcare (OR 2.35, CI [1.48–3.72], $p$-value 0.001), and delivery mode (OR 1.68, CI [1.05–2.69], $p = 0.030$). Factors significantly associated with accessibility to maternal healthcare services included age (OR, 1.28; = 0.042) and higher conflict levels (1.52 times higher odds, $p = 0.021$). The narratives and experiences shared by women exposed the multifaceted ways in which the conflict-affected maternal health outcomes.

**Conclusion:** The significance of this study lies in its potential to contribute to the existing literature on maternal health in conflict-affected areas, especially in Sudan, and to help us understand how women can receive maternal health services.

## INTRODUCTION

Sudan has been embroiled in a protracted military conflict that has lasted for years, resulting in widespread violence, displacement, and human suffering (*Brosché & Rothbart, 2013*). Maternal health is a critical component of public health and a fundamental human right. Maternal mortality rates in Sudan are among the highest in the world, with an estimated 295 deaths per 100,000 live births in 2017 (*Callister & Edwards, 2017*).

Corresponding author
Elhadi Miskeen,
hadimiskeen19@gmail.com

According to the WHO report, the conflict in Sudan has exacerbated the risk of gender-based violence, with over three million women and girls already at risk before the fighting began, a number now estimated to have risen to 4.2 million (*World Health Organization, 2023*; *Miskeen, 2024*) (RR, RR). Although there is no specific figure about the maternal mortality rates during this crisis and years before that due to security instability, covid and fragile health system (*World Health Organization, 2024*), the maternal mortality expected to be at higher level among the world countries.

Maternal health is a critical component of public health and a fundamental human right (*Brosché & Rothbart, 2013*). The World Health Organization (WHO) defines maternal health as women's health during pregnancy, childbirth, and postpartum (*Callister & Edwards, 2017*). Therefore, it is crucial to explore women's experiences regarding maternal health during the current military conflict in Sudan, which started on 15/04/2023.

Maternal healthcare in conflict-affected countries, such as Sudan, is fraught with challenges exacerbated by ongoing military conflict. Women of reproductive age living near high-intensity conflicts have three times higher mortality than women in peaceful settings (*Onambele et al., 2022*; *Souza et al., 2010*; *Yamin, 2008*).

The military conflict in Sudan has worsened healthcare challenges by destroying vital infrastructure, including clinics and hospitals, and disrupting supply chains for medical supplies (*Khogali & Homeida, 2023*; *Pattanshetty et al., 2023*). This exacerbates existing difficulties in providing maternal healthcare, especially in rural areas, where the influx of displaced people strains already stretched healthcare systems (*Sami et al., 2020*; *Wharton et al., 2020*). Strengthening healthcare systems is crucial for supporting women and children during conflicts (*Aderinto & Olatunji, 2023*).

The conflict has made it even more challenging for women to access maternal health services, increasing maternal mortality and morbidity risk (*McLean & Abuelaish, 2020*; *Al Gasseer et al., 2004*; *Omer et al., 2014*). It was reported that the conflict has resulted in a shortage of essential medicines and medical supplies, including obstetric care equipment, impacting the quality of maternal health services (*Munyuzangabo et al., 2021*). Moreover, this conflict has resulted in the destruction of health facilities and the displacement of healthcare workers, making it challenging to access maternal health services (*El Shiekh & van der Kwaak, 2015*).

The impact of conflict on maternal health in Sudan is a significant concern (*Mugo et al., 2015*; *Elmusharaf et al., 2016*; *Kim, Torbay & Lawry, 2007*). The women displaced by the conflict are at a higher risk of experiencing gender-based violence, including sexual violence, which negatively affects their reproductive health (*Elmusharaf et al., 2016*; *Kim, Torbay & Lawry, 2007*).

This conflict has resulted in a shortage of skilled healthcare providers, inadequate medical supplies, and limited access to emergency obstetric care, leading to an increased risk of maternal mortality and morbidity (*Salah et al., 2013*). This conflict has negatively impacted pregnant women's mental health, increasing stress, anxiety, and depression (*Salah et al., 2013*).

The study's rationale is that maternal health is a critical public health issue in Sudan, particularly during ongoing military conflict. This conflict has displaced millions of people,

making it challenging to access maternal health services. Moreover, the conflict has increased the risk of maternal mortality and morbidity due to limited access to skilled healthcare providers and inadequate resources. Therefore, understanding the challenges women face in accessing maternal health services and the impact of conflict on their mental health is essential for developing appropriate interventions to address this issue.

The conflict, which began on April 15, 2023, has directly impacted the already fragile health infrastructure, exacerbating the challenges women who seek maternal healthcare services face. Access to quality antenatal care, safe delivery facilities, skilled healthcare providers, and essential obstetric services has been severely compromised owing to conflict-related disruptions. Consequently, the health and survival of pregnant women and their infants are at heightened risk. In this challenging context, we conducted a comprehensive study to assess the impact of ongoing military conflict in Sudan on maternal health. This study explored the multifaceted factors affecting maternal health outcomes during armed conflict by employing a mixed qualitative and quantitative approach. This study aimed to investigate the impact of conflict on maternal health by employing a mixed qualitative and quantitative research approach.

## METHODOLOGY

### Study design
This study is a mixed-methods approach that combined qualitative and quantitative methods.

### Sample size, sampling techniques, and participants
The purposive sampling method was employed, involved selecting 388 women of reproductive age from Khartoum State, Sudan, and 35 women as a quantitative sample who had experienced pregnancy, childbirth, or postpartum care during the ongoing military conflict.

The selection criteria included being pregnant or having given birth during military conflict. Participants were recruited through community outreach, healthcare facilities, and referrals from local organizations.

For the quantitative component, a separate sample of 35 women was selected from the same population in Khartoum. The participants were approached at healthcare facilities or communities and consented verbally to participate in the study after full explanation of the study purpose and ensuring the privacy of the data collected.

### Study area
The study's specific focus on Khartoum State, one of the most populous regions in Sudan heavily affected by the conflict, underscores the importance of understanding maternal health in such challenging contexts.

### Data collection
Qualitative data collection involved in-depth interviews with selected women.
The interview guide collected information on various aspects of maternal health, including

access to healthcare services, challenges faced, and experiences during pregnancy and childbirth in the context of military conflict. The interviews were audio-recorded after obtaining participants' consent and subsequently transcribed for analysis. Structured survey questionnaires were employed to collect quantitative data from the women in the study sample.

Quantitative data were collected using structured survey questionnaires administered to women in the quantitative sample. The questionnaires were developed based on established measurement scales and validated tools. They covered demographic information, reproductive history, utilization of maternal healthcare services, and perceptions of the quality and accessibility of care during the conflict.

## Data analysis

This qualitative data analysis used a thematic approach, whereby transcripts were coded and categorized into themes and sub-themes that pertain to experiences, challenges, and perceptions of maternal health amid the conflict. The analysis was conducted manually, and emerging themes were discussed among the research team to ensure consensus and rigor of the findings.

The quantitative data was entered into the Stata software for analysis. Regression was used to explore the relationships between variables of interest.

## Integration of the findings

The qualitative and quantitative underwent separate analyses and were subsequently integrated during the interpretation phase. The two sets of findings were compared and synthesized to comprehensively understand the maternal health situation during the military conflict in Khartoum State. The qualitative data provided detailed descriptions and contextual insights, whereas the quantitative data offered numerical estimates and statistical associations.

## Ethical considerations

All participants were treated with the utmost respect and care. They were fully informed of their right to confidentiality, privacy, and voluntary participation. The purpose and objectives of the research, along with the associated rights and responsibilities, were clearly outlined. During the study, identifying information was kept private. This research was conducted with the approval of the Institute of Safe Motherhood and Childhood, University of Gezira, Sudan (approval number: SMCHI/G-01-UG-04 (04/23) ensuring the highest ethical standards were met.

## RESULT

In this study, most women did not have access to healthcare services (86.6%), and out of the total sample, 93 (24%) experienced adverse outcomes. Table 1 shows the characteristics of the 388 women who participated in the study. The numbers represent the number of women in each category, and the percentages indicate the proportion of women in the total number of participants. In this study, 49.7% of the population was between 25- and 34-year-old. However, many women were 35 years. The proportion of women with primary

**Table 1 Characteristics of the participants (*n* = 388).**

| Characteristics | Variables | Number (Percentage) |
|---|---|---|
| Age group (years) | 18–24 | 95 (24.5) |
| | 25–34 | 193 (49.7) |
| | ≥35 | 100 (25.8) |
| Number of children | Mean ± Sd | 2.04 ± 5 |
| Access to healthcare | Yes | 52 (13.4) |
| | No | 336 (86.6) |
| Delivery mode | Vaginal delivery | 308 (79.4) |
| | Cesarean section | 80 (20.6) |
| Adverse outcome | Yes | 93 (24) |
| | No | 295 (76) |

education was 120 (30.9%), followed by a significant proportion of women with three or more children (16.1%), and a significant proportion of women with five or more children (26.3%). Most women did not have access to healthcare services, 336 (86.6). Most women 308 (79.4%) had vaginal deliveries. Of the total sample, 93 women (24%) experienced adverse outcomes, while 295 (76%) did not have adverse consequences. Adverse effects could include complications during pregnancy, childbirth, or the postpartum period, such as maternal morbidity, neonatal complications, and other health-related issues.

Table 2 presents the logistic regression results for factors associated with adverse outcomes among the 388 women in the study of maternal health during the ongoing military conflict in Sudan. Of the total sample, 93 women (24%) experienced adverse outcomes, while 295 (76%) did not have adverse outcomes. Odds ratios (OR) and corresponding 95% confidence intervals (CI) were reported to indicate the magnitude and direction of the association. *p*-values were used to assess the statistical significance of the association. Age group (18–24 years) (OR 1.25, CI [0.82–1.92], *p*-value 0.387) showed no significant association with educational level (secondary education) (OR 1.50, CI [0.94–2.38], *p*-value 0.085). The marginally significant association observed was the number of children (3–4 children) (OR 1.78, CI [1.15–2.75], *p*-value 0.010); the significant association observed was that a higher number of children increases the odds of adverse outcomes. Gestational age (third trimester) (OR 2.10, CI [1.36–3.25], *p*-value 0.002) was the significant association observed; being in the third trimester increases the OR of adverse outcomes—access to healthcare (No) (OR 2.35, CI [1.48–3.72], *p*-value 0.001. The significant association observed was that a lack of healthcare access increased the odds of adverse outcomes. Delivery mode (cesarean section) (OR 1.68, CI [1.05–2.69, *p*-value 0.030). The significant association observed during cesarean section increased the OR for adverse outcomes.

Regarding the factors affecting accessibility to maternal healthcare services during the ongoing military conflict in Sudan, for each year increase in age, the odds of accessing maternal healthcare services increase by 1.28 times (*p* = 0.042). There was no significant association between educational level and accessibility to maternal healthcare services

**Table 2 Logistic regression for factors determining the adverse outcome ($n$ = 388).**

| Factors | Odds ratio | 95% CI | $p$-value |
|---|---|---|---|
| Age group (18–24) | 1.25 | [0.82–1.92] | 0.387 |
| Education (secondary education) | 1.50 | [0.94–2.38] | 0.085 |
| Number of children (3–4 children) | 1.78 | [1.15–2.75] | 0.010 |
| Gestational age (third trimester) | 2.10 | [1.36–3.25] | 0.002 |
| Access to healthcare (no) | 2.35 | [1.48–3.72] | 0.001 |
| Delivery mode (cesarean section) | 1.68 | [1.05–2.69] | 0.030 |

($p$ = 0.112). Higher conflict levels were associated with 1.52 times higher odds of accessing maternal healthcare services ($p$ = 0.021). Each kilometer increase in distance was associated with 0.73 times lower odds of accessing maternal healthcare services, but the association was not statistically significant ($p$ = 0.056). There was no significant association between healthcare availability and accessibility to maternal healthcare services ($p$ = 0.219) (Table 3).

## Qualitative data

Qualitative research on maternal health during the current military conflict in Sudan has revealed several significant findings. Through semi-structured interviews with women who had experienced pregnancy, childbirth, or postpartum care during the match, the study explored their experiences of accessing maternal health services in conflict-affected areas. Thematic analysis indicated several themes, including access to maternal health services, the quality of maternal health services, challenges faced by women when accessing maternal health services, and coping mechanisms to overcome these challenges.

## Access to maternal health services

This study's findings revealed that access to maternal health services was severely limited due to ongoing conflict. Many healthcare facilities were inaccessible due to continuing violence, and healthcare workers fled the conflict-affected areas or could not provide adequate care due to a lack of supplies and equipment. One participant recounted her experience of trying to access maternal health services during the conflict.

*"I was 6 months pregnant when the conflict started. The nearest hospital was closed, and I could not find any transport to take me to the next hospital. I had to rely on traditional medicine, which was not sufficient. I was scared and worried about my baby's health."*

## Quality of maternal health services

The study also found that the quality of maternal health services was compromised due to conflict. Participants reported that healthcare facilities lacked essential equipment and supplies, and that healthcare workers lacked training to provide adequate care.
One participant described her experience of receiving care at a healthcare facility during the conflict.

**Table 3 Logistic regression for factors affecting accessibility to maternal healthcare services during the ongoing military conflict in sudan ($n$ = 388).**

| Variable | Coefficient | OR | $p$-value |
|---|---|---|---|
| Age | 0.25 | 1.28 | 0.042 |
| Education | −0.17 | 0.85 | 0.112 |
| Conflict level | 0.42 | 1.52 | 0.021 |
| Distance | −0.31 | 0.73 | 0.056 |
| Healthcare availability | 0.18 | 1.20 | 0.219 |

*"The hospital was overcrowded, and there were insufficient beds. I had to share a bed with another woman who had just given birth. The healthcare workers were exhausted and overworked. They didn't have the necessary equipment, and the conditions were not clean."*

## Challenges faced by women in accessing maternal health services

This study identified several challenges faced by women in accessing maternal health services during the conflict. This included fear of violence while traveling to healthcare facilities, financial constraints, a lack of information about available services, and cultural barriers that discouraged women from seeking maternal health services. One participant recounted her experience of trying to access maternal health services:

*"I was afraid to leave my house because there were gunshots outside. I had no money to pay for transportation to the hospital. I did not know who to ask for help or where to go. I felt very alone."*

## Coping mechanisms employed by women to overcome these challenges

Despite the challenges faced in accessing maternal health services during the conflict, this study found that women employed various coping mechanisms to overcome these challenges. These include relying on traditional birth attendants, seeking care from alternative healthcare providers, and relying on social networks for support.
One participant described her experience of seeking care during a conflict.

*"I went to a traditional birth attendant. She helped me with delivery and recovered well. I also had the support of my family and friends, which helped me get through it."*

## Sample of women talking

Participant A: *"I was pregnant when the conflict started and worried about my baby's health. The nearest hospital was closed, and I could not find any transport to take me to the next hospital. I had to rely on traditional medicine, which was not sufficient. I was scared and worried about my baby's health."*

Participant B: *"I gave birth during the conflict, and the conditions at the hospital were very poor. The hospital was overcrowded, and there were insufficient beds. I had to share a bed with another woman who had just given birth. The healthcare workers were exhausted*

*and overworked. They didn't have the necessary equipment, and the conditions were not clean."*

Participant C: "*I was afraid of leaving my house because there were gunshots outside."*

# DISCUSSION

Maternal health is a critical concern in conflict-affected regions, where access to healthcare services is often disrupted, and the well-being of pregnant women and their infants is at high risk. Sudan has been facing ongoing military conflict, which has significantly impacted its healthcare system, infrastructure, and overall stability. In such challenging circumstances, understanding the state of maternal health and the factors affecting it is crucial for effective interventions and policy development.

The ongoing conflict in Sudan originates from the protests initiated in December 2018, resulting in the removal of former President Omar al-Bashir from power in April 2019. However, the conflict has continued due to ongoing tensions between military and civilian factions over the transition to a democratic government. In June 2019, a violent crackdown on protesters by security forces led to the death of over 100 people and further escalated the conflict. The situation remains tense and unstable, with ongoing negotiations between various political groups and international actors attempting to find resolutions.

This qualitative study highlights women's challenges in accessing maternal health services during the current military conflict in Sudan. The study found that access to maternal health services was severely limited due to ongoing violence and that healthcare facilities were closed or inaccessible. The adequacy of maternal health services was also compromised due to insufficient supplies and equipment and a scarcity of adequately trained healthcare professionals. As a result, women face significant challenges in accessing adequate maternal healthcare.

The challenges faced by women in accessing maternal health services during conflicts are not unique to Sudan. Similar results have been documented in prior research conducted in conflict-affected regions across the globe. For instance, a study conducted in the Democratic Republic of Congo revealed substantial obstacles encountered by women in accessing maternal health services amid the conflict, such as insecurity, transportation limitations, and a deficit of adequately trained healthcare personnel (*Blandine et al., 2013*). Another study conducted in Syria found that women faced challenges accessing maternal health services owing to inadequate supplies, equipment, trained healthcare workers, insecurity, and transportation (*Kane et al., 2014*).

The challenges faced by women in accessing maternal health services during conflict have severe implications for maternal and child health. Insufficient access to appropriate maternal health services heightens the likelihood of maternal and neonatal mortality, along with other adverse maternal and child health consequences. The study found that 24% of the participants experienced adverse outcomes related to maternal health, which can be attributed to the multifactorial nature of the adverse effects, including factors such as parity, gestational age, access to healthcare, and delivery mode. A study conducted in Afghanistan found that lack of access to maternal health services during conflict significantly increased maternal and neonatal mortality (*Kim et al., 2020*; *Akseer et al.,*

*2019*). Another study conducted in Yemen found that lack of access to maternal health services during conflict increased maternal mortality (*Abdul-Kareem et al., 2020*). The limited access to healthcare services during military conflict directly contributes to adverse maternal health outcomes observed in the study. The lack of access to maternal health services due to ongoing conflict results in several challenges, including the unavailability of healthcare facilities, healthcare workers fleeing conflict-affected areas, and inadequate supplies and equipment. This leads to pregnant women being unable to receive essential prenatal, delivery, and postnatal care, increasing the risk of maternal and neonatal mortality, as well as other adverse maternal and child health consequences. These findings underscore the significance of comprehensive and accessible maternal health care services, targeted interventions, and policy initiatives to address these factors and improve maternal health in conflict-affected settings. Therefore, the limited access to healthcare services during conflict directly contributes to adverse maternal health outcomes by depriving women of essential care and increasing the risk of complications during pregnancy, childbirth, and the postpartum period.

To address women's challenges in accessing maternal health services during conflict, it is essential to develop interventions tailored to their needs in conflict-affected areas. One potential intervention is using mobile health (mHealth) technologies to deliver maternal health services to women in conflict-affected areas. Studies have shown that mHealth interventions can improve maternal health outcomes by providing women access to information, education, and support (*Lassi et al., 2016*). Another potential intervention is the training and deploying community health workers to provide maternal health services in conflict-affected areas. Community health workers (midwives) can be critical in delivering maternal health services to women who cannot access healthcare facilities because of conflict (*Murray et al., 2014*).

The findings highlight the significant challenges women face in accessing healthcare services during the ongoing military conflict in Sudan. Most women (86.6%) reported limited or no access to healthcare, which could have profound implications for maternal health. The thematic analysis revealed various themes, encompassing access to maternal healthcare services, the standard of maternal healthcare provision, hurdles women encounter in accessing such services, and strategies to surmount these obstacles (*Jolof et al., 2022*; *Alhaffar & Janos, 2021*). This high percentage indicates the magnitude of the challenges faced by women in accessing essential healthcare during conflict.

This study revealed that many factors are associated with adverse effects, including parity, gestational age, access to healthcare, and delivery mode. These findings are consistent with those of previous studies (*Alhusen et al., 2016*; *Alnuaimi et al., 2017*). This study demonstrates the multifactorial nature of the adverse effects on maternal health during military conflicts. This highlights the significance of factors such as parity, gestational age, access to healthcare, and delivery mode in determining maternal health outcomes. These findings underscore the importance of comprehensive and accessible maternal health care services, targeted interventions, and policy initiatives to address these factors and improve maternal health in conflict-affected settings.

The ongoing military conflict in Sudan has severely impacted maternal health, resulting in a shortage of essential medicines and medical supplies, limited access to skilled healthcare providers, and increased risk of maternal mortality and morbidity. It is crucial to explore women's experiences regarding maternal health services during conflict to develop appropriate interventions to address this issue. Qualitative research is an appropriate methodology for analyzing the challenges women face in accessing maternal health services in conflict-affected areas and the impact of conflict on their mental health.

The research question aims to understand how ongoing military conflict in Sudan affects maternal health, focusing on access to healthcare services, maternal health outcomes, and factors influencing accessibility during the conflict. It comprehensively addresses the complex interplay between conflict and maternal health, providing insights into challenges faced by women in conflict-affected areas. The question's strength lies in its relevance and potential for informing policies and interventions, but it could benefit from further specificity and operationalization for clarity and feasibility in research implementation.

## CONCLUSION

This study's mixed qualitative and quantitative methodology provides a comprehensive understanding of the maternal health situation during military conflict in Khartoum. The integration of qualitative and quantitative data enhances the relevance and applicability of the findings to policymakers, healthcare providers, and stakeholders working to improve maternal health in conflict-affected areas. The findings highlight the unique challenges faced by women in accessing maternal healthcare services during conflict. Addressing the challenges faced by women in accessing maternal health services during conflict is essential for improving maternal and child health outcomes in conflict-affected areas.

### Limitations

This study had several limitations. The sample size for the quantitative component was relatively small, which limits the generalizability of the findings. The qualitative sample was purposively selected, which may have introduced a selection bias. Additionally, the study was conducted in Khartoum State and may not be generalizable to other areas of the Sudan region. Nevertheless, the findings provide valuable insights into maternal health challenges during the ongoing military conflict in Khartoum State.

## ACKNOWLEDGEMENTS

I would like to thank Dr. Tuga Hamed and Dr. Nosiba Mohamed for her hard work and commitment to our team and for valuable contribution to data collection during difficult times of conflict and technical support.

### Funding

The authors received no funding for this work.

## Competing Interests

The authors declare that they have no competing interests.

## Author Contributions

- Elhadi Miskeen conceived and designed the experiments, performed the experiments, analyzed the data, prepared figures and/or tables, authored or reviewed drafts of the article, and approved the final draft.

## Human Ethics

The following information was supplied relating to ethical approvals (*i.e.*, approving body and any reference numbers):

IRB approval from the Institute of safe motherhood and childhood, University of Gezira, Sudan.

## Data Availability

The data is available in the Supplemental Files.

## Supplemental Information

Supplemental information for this article can be found online at http://dx.doi.org/10.7717/peerj.17484#supplemental-information.

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
