# Peer review of "The impact of the military conflict in Sudan on maternal health: a mixed qualitative and quantitative study"

_PeerJ, doi:10.7717/peerj.17484_

## Round 0.1 · original submission · Major Revisions

The manuscript underwent a thorough review by two reviewers who provided constructive feedback on various aspects of the study.

Reviewer 1 noted that while the manuscript effectively links the theoretical framework to the research problem and cites relevant literature, there is a need for more clarity and detail in explaining the sampling techniques. The reviewer also raised concerns about the use of parametric statistics with nominal data. Additionally, the discussion was found to be somewhat disconnected from the research questions.

Reviewer 3 pointed out language and coherence issues in the manuscript, suggesting that editing is needed for improved clarity. There was a request for specific details on the mixed-method design used and explicit explanations of the sampling techniques. The reviewer also highlighted concerns regarding missing information in Table 1 and the discussion's alignment with the study's aim.

Both reviewers suggested revisions to address these issues and enhance the manuscript's overall quality.

**Language Note:** The Academic Editor has identified that the English language must be improved. PeerJ can provide language editing services - please contact us at [email protected] for pricing (be sure to provide your manuscript number and title). Alternatively, you should make your own arrangements to improve the language quality and provide details in your response letter. – PeerJ Staff

Reviewer 1 ·

Basic reporting

The theoretical framework is linked to the research problem. identifies relevant research and literature and summarizes and integrates the information. Cites works and places them in context. Research questions are clearly articulated and sufficient background information. Shows knowledge of methodology and gives justification for the selection of methods. however, sampling techniques need more clarity and detailing. Results are clearly summarized, discussion of results-focused and tied to the research question.

Experimental design

The sample of the study was selected purposively and the quantitative data level is nominal but parametric statistics was employed which seems contradictory.

Validity of the findings

Discussion of results less connected to research questions. The new knowledge gained from the study needs to be more clarity.

Additional comments

Average organization of ideas, and writing hinders reader understanding. More connectedness was expected.

·

Basic reporting

No comment

Experimental design

No comment

Validity of the findings

No comment

·

Basic reporting

The research as an interesting focus but the use of English Language, coherence and flow can be improved through editing for example line 32 sentence 2 and line 43 sentence 1.

Experimental design

Please state the specific type of mixed-method design adopted for the study. I believe the author used a technique similar to Mixed Method Triangulation Design.

The author needs to explicitly explain the specific sampling techniques used and how it was incorporated to aid better understanding for replication purposes.

A justification citation may be needed in line 71 sentence 1.

In line 78 please state the socio-demographic components catered for in selection.

Validity of the findings

It was not stated clearly if participants signed consent forms either digitally or physically.

In table 1, some information described are missing, such as education.

The discussion part has not comprehensively addressed the aim of the study and attributed components.

The consent form did not request approval to participate but approval to be brief on the study.

The questionnaire is not rigorous.

---

## Round 0.2 · Minor Revisions

Based on the provided information, it appears that the manuscript meets the basic reporting criteria. The research questions are clearly articulated, and there is sufficient background information included. The discussion of results is focused and connected to the research questions, with implications for future research discussed.

In terms of experimental design, concerns about the methodology were incorporated, suggesting a thorough review process.

The validity of the findings is supported by the discussion of results, which is focused and connected to the research questions.

Overall, the manuscript appears to be well-structured and addresses the key aspects of the review criteria. Therefore, the editor decision would likely be to accept the manuscript, pending minor revisions based on reviewer comments.

Reviewer 1 ·

Basic reporting

Research questions are articulated and sufficient background information is included.
Discussion of results-focused and connected to research questions. Implications for future research
discussed.

Experimental design

concerns about the methodology were incorporated.

Validity of the findings

Discussion of results-focused and connected to research questions

Additional comments

Results were interpreted in light of the proposed research question and existing literature.

·

Basic reporting

Generally, the introduction has clarity and needs editing but can be improved with better flow and coherence. The author can start with the conflict in Sudan and then connect with maternal health and how it has been affected. Also, adverse outcomes are prominent in the quantitative result, the author may consider including it in the introduction. I am suggesting paragraphs 1 (lines 33-37) and 2 (lines 38-43) to be switched.
Lines 33 and 40 are a repetition, I suggest more literature search is done to make it robust.
Clarify the competition you are referring to in line 39.
It would be interesting to get an update (within 5 years) on statistics on maternal death as mentioned in line 43.
Please indicate the study you are referring to in line 49
The author may want to clearly state the objectives to guide the reader towards the results. There seems to be a little disparity.

Experimental design

There is a mix up with the participant number and study method in lines 86 and line 92. Please correct with information provided by you line 76.
Expand on how recruiting from these sources amount to diversity in demographics as mentioned in lines 88-89.
Indicate if the consent was in verbal or written form as mentioned in line 93-94.
Please expand on why and how participants for the qualitative study randomized as mentioned in line 79.
The questionnaire is not robust and does it indicate variables of interest. This has influenced the quantitative result, the need for regression was not paramount as variables were interacted without focus DV

Validity of the findings

The results are clear and useful but not tied to an objective, outcomes should be associated with the uneasy access to health care plus its influence on adverse outcomes due to the conflict

---

## Round 0.3 · accepted · Accept

The author has revised the manuscript according to the reviewer's feedback, ensuring that all changes have been made to improve overall quality of the content. Please proceed with the publication process for this manuscript.